# Predicting real-world navigation performance from a virtual navigation task in older adults

Sarah Goodroe[1], Pablo Fernandez Velasco[2,3]*, Christoffer J Gahnstrom[1], Jan Wiener[4], Antoine Coutrot[5], Michael Hornberger[6], Hugo J Spiers[2]

1 Department of Psychology, University of Pennsylvania, Pennsylvania, United States of America, 2 Institute of Behavioural Neuroscience, Department of Experimental Psychology, Division of Psychology and Language Sciences, University College London, London, United Kingdom, 3 Department of Philosophy, University of York, York, United Kingdom, 4 Department of Psychology, Ageing and Dementia Research Centre, Bournemouth University, Poole, United Kingdom, 5 LIRIS, CNRS, University of Lyon, Lyon, France, 6 Norwich Medical School, University of East Anglia, Norwich, United Kingdom

☉ These authors contributed equally to this work.
* pablo.velasco.18@ucl.ac.uk

Virtual reality environments presented on tablets and smartphones offer a novel way of measuring navigation skill and predicting real-world navigation problems. The extent to which such virtual tests are effective at predicting navigation in older populations remains unclear. We compared the performance of 20 older participants (54–74 years old) in wayfinding tasks in a real-world environment in London, UK, and in similar tasks designed in a mobile app-based test of navigation (Sea Hero Quest). In a previous study with young participants (18–35 years old), we were able to predict navigation performance in real-world tasks in London and Paris using this mobile app. We find that for the older cohort, virtual navigation performance predicts real-world performance for medium difficulty, but not for the easy or difficult environments. Overall, our study supports the utility of using digital tests of spatial cognition in older age groups, while carefully adapting the task difficulty to the population.

## Introduction

Digital tools and video games offer a substantial expansion in the traditional toolkit of psychologists and cognitive scientists [1,2]. Video games tend to be enjoyable, and thus able to sustain participants' engagement, motivation and attention [3]. Moreover, the digital medium offers a unique avenue for crowdsourcing and big data approaches [4,5]. Finally, virtual environments can be designed to resemble real-world settings as a way of increasing the ecological validity of the experimental task [6]. A case in point is Sea Hero Quest, a mobile game app which simulates a nautical themed navigation adventure collecting virtual trajectory data and participant demographics [7,8]. It was designed to test and benchmark spatial cognition world-wide. Since its release, more than 4 million participants across every nation-state have downloaded and played Sea Hero Quest, leading to a series of transnational studies across psychology and the cognitive sciences [8].

While virtual environments provide control and easily reproducible experiments, it is important that they provide ecological validity both for helping understanding cognition

**Data availability statement:** The data supporting the findings of this study are available as open data via the following online data repository link: https://osf.io/eq65d/?view_only=e-531fa874e70474ab7b7ad3775e4ed62

**Funding:** This research is part of the Sea Hero Quest initiative funded and supported by Deutsche Telekom. Alzheimer's Research UK (ARUK-DT2016-1) funded support for the research; Glitchers designed and produced the game; and Saatchi and Saatchi London managed its creation. The funders had no role in study design, data collection and analysis, decision to publish, or preparation of the manuscript.

**Competing interests:** The authors have declared that no competing interests exist.

in the real-world, but also for valid assessment of neuropsychology [9]. Numerous studies in recent years have found evidence for the increased ecological validity of virtual reality in spatial research both for healthy individuals [10–14] and for those with Alzheimer's disease (AD) or pre-AD [15–17]. This is in contrast to studies using object-based spatial tasks (e.g., the Corsi task to test visuospatial working memory), which have mixed results in predicting real-world behaviour (e.g., path integration, wayfinding) [18–21].

The detail of data (time-series coordinate data) and the volume of the data originating from Sea Hero Quest allows for fine-grained analysis of the interactions of sociodemographic and geographic factors with navigation ability [22]. Studies using Sea Hero Quest have found that gender inequality across countries (gender gap index) predicts gender differences in navigation ability [7], that 7 hours of sleep is associated with a better navigation performance late in life across cultures [23,24], or that neither handedness [25] nor GPS reliance [26] are associated with navigation ability, while education is [27]. Cultural norms around masculinity have been found associated with self-over-estimation of navigation performance across countries [28]. Sea Hero Quest has also shown that the environment in which one grows up shapes navigation skills later in life. Measuring the characteristic entropy of city street networks of different countries and contrasting this with Sea Hero Quest data revealed that people who grew up in griddy cities (e.g., Toronto) were better at navigating griddy environments, with the opposite effect for people who grew up in rural areas or non-griddy cities [8,29]. All of these discoveries depend on the ecological validity of Sea Hero Quest to be able to relate the findings to real-world behaviour.

In a previous study, Sea Hero Quest was found to be a reliable predictor of real-world navigation ability in healthy, young populations both in London and in Paris, suggesting that the game is an ecologically valid measure of wayfinding ability [30]. In that previous study, we compared the performance in a subset of wayfinding levels in Sea Hero Quest with the performance in a wayfinding task in the area of Covent Garden, London UK, and found a significant correlation between the distance participants travelled in the video game, measured in pixels, and in the street network, measured in metres in the real-world through a GPS device. We then replicated the result with another set of participants in the area of Montparnasse in Paris. Whilst this is important to show ecological validity of an app-based assessment, the extent to which it could predict real-world navigation in later life is unclear due to the decline in navigation performance that occurs with age [7,31–37]. Furthermore, a significant value in developing tests of navigation is that they may be useful in the early diagnosis and cognitive monitoring of Alzheimer's disease. Alzhiemer's disease or other neurological conditions [16,17,22,38–40]. Deficits in navigation have higher specificity than episodic memory when it comes to distinguishing Alzheimer's disease from other dementias. Animal studies have also shown that Alzheimer's disease pathophysiology affects brain areas associated with navigation before brain areas associated with episodic memory [38].

The aim of the present study was to test whether Sea Hero Quest was predictive of real-world navigation performance in an older population. To this end, we tested an older cohort both in Sea Hero Quest tasks and in real-world wayfinding tasks in London, following the same paradigm as the younger cohort in our previous study [30]. We hypothesised that performance in the real world would significantly correlate with performance in the virtual environment. Moreover, we expected to find a general decline reflected in a difference between this older group and the younger group in our previous study, as the age-related decline in navigation ability is a robust effect seen across studies [7,31–37].

## Methods

This study was approved by University College London Ethics Research Committee. The ethics project ID number is CPB/2013/015. Written consent was obtained from each participant.

The data was analysed anonymously. For the mobile gameplay component of the task, participants were tested on a specific subset of Sea Hero Quest levels [30] on either an Acer tablet or an iPad. For the real-world component, participants were tested on a similar task in the streets of Covent Garden, London, UK. Participants were asked to answer demographic questions, and they completed the Navigation Strategy Questionnaire (NSQ) [41,42]. This experiment was conducted in London during spring and summer of 2019 (01.06–30.09). The total testing time per session lasted around three hours per participant.

## Participants

We tested a total of 24 participants (6 males), recruited through Alzheimer's Research UK, University of the Third Age, and AgeUK. The task necessitated walking during an extended period, so we collected as many participants as possible given this practical constraint and the project duration. Our aim was to collect a sample size that matched that of the previous study with a younger cohort, and we fulfilled that aim. Data from three participants (3 female) was excluded from group analysis due to GPS tracking issues during the real-world component of the task. Data from one participant (male) was excluded from group analysis due to non-completion of the mobile gameplay component. The subsequent analysis represents that of the remaining 20 participants (5 males) (mean age = 68.2 ± 4.23 years, range = 54–74). All participants had normal or corrected to normal vision, confirmed that they were able to comfortably walk for around two hours, and gave their written consent to participate in the protocol as outlined in an informational sheet read before consent was obtained. Participants were compensated for their time with £25, as well as tea and biscuits, for their participation. To control for general cognitive decline with ageing, we asked participants to complete the Montreal Cognitive Assessment before starting the main navigation task [43]. We implemented the threshold of 26 points out of 30 which has a reported rate of detection of 90% and 100% for MCI and mild AD, respectively [43]. One participant was excluded from further participation due to a total score of below 26. For the 20 participants included in this analysis, all scored between 26 and 30 (mean score = 28.45 ± 1.2).

## Virtual tasks

We designed a mobile video game, Sea Hero Quest, to measure spatial navigation ability. Participants navigate a boat in a virtual environment, using either an Acer tablet or an iPad as an interface (for an extensive description, see [7]). At the start of each level, participants viewed a map displaying their current position and ordered goal locations in navigable waterways of the environment. In this task, participants were able to study the map without any time restriction. When done studying, the participants pressed play and then (from a view that positionied the player's perspective behind the boat) had to navigate to the goal locations in the order indicated on the map by number (i.e., find goal 1, find goal 2, … find goal n, until all n goals have been located), see Fig 1A,E. In the virtual water network, goals corresponded to buoys with flags signalling the goal number, with a circle around the buoy indicating the goal area. Participants had to enter the corresponding circle to successfully reach each goal. Once all goals were reached, the level was registered as complete, and participants moved on to the next level, until all six levels were completed. For each level, a guidance arrow appears on the screen after a certain duration, indicating the direction to the goal along the vector to the goal. This arrow appeared after the following durations by level: 80s in Level 1, 70s in Level 6, 80s in Level 11, 80s in Level 16, and 200s in Level 43.

Based on several pilot sessions, we decided to include a brief training on the game controls before participants played the specified levels of Sea Hero Quest. Level 1 of Sea Hero Quest indicates the video game controls (e.g., tap to steer left or right). Following pilot work, and

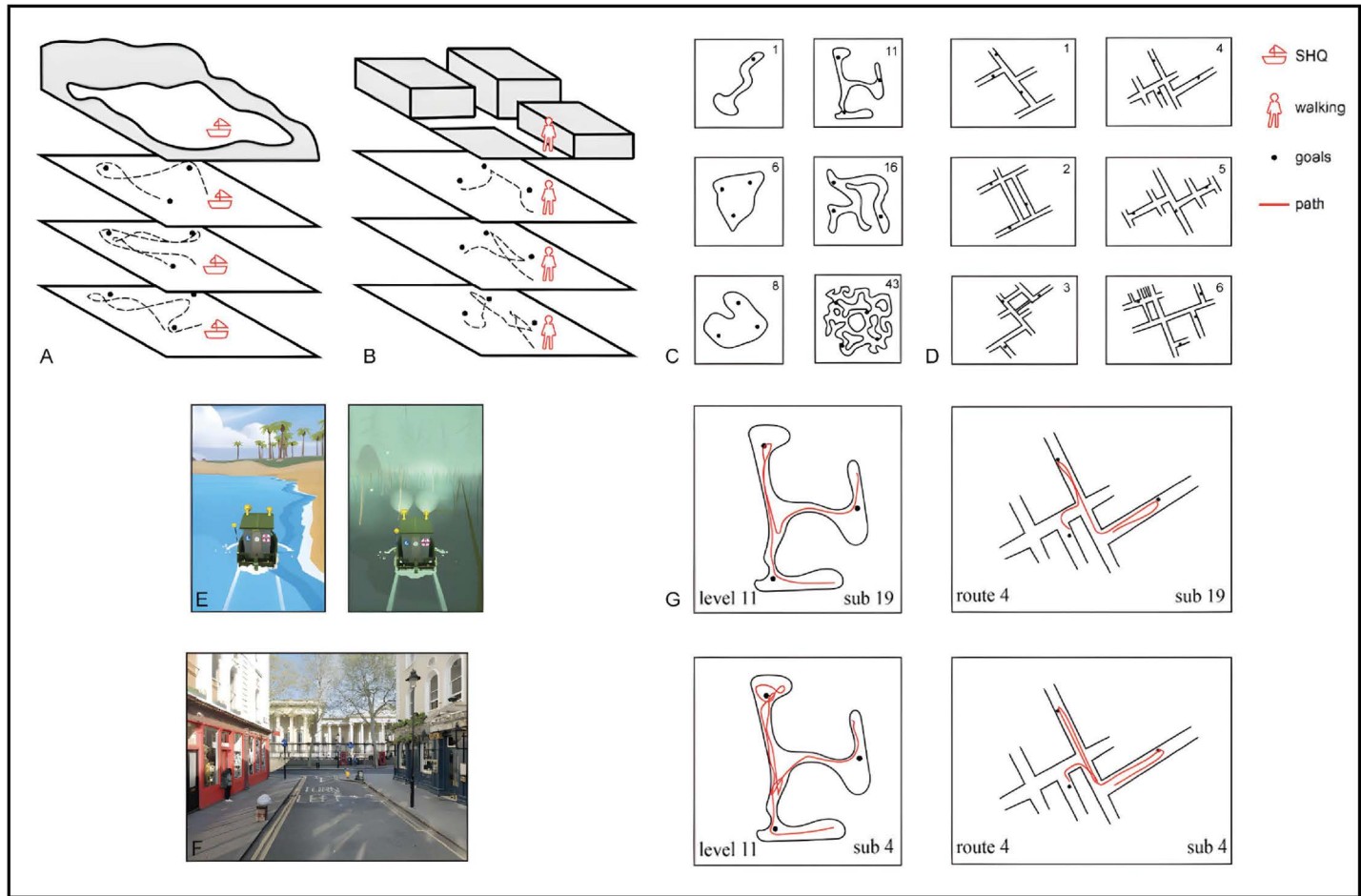

**Fig 1. SHQ and real-world wayfinding tasks.** (A) Example trajectories within Sea Hero Quest, in which participants' movement was restricted only by the walls within the water environment. (B) Example trajectories in the real-world wayfinding task, in which participants' movement was restricted to the pavement and also experimenter-induced barriers around the testable space. (C) Six levels of Sea Hero Quest tested: 1, 6, 8, 11, 16, & 43 (level 1 functioned as a training level + measure of video game skill). Note level 8 was not tested for both older and younger participants and is thus not included in our analysis. (D) Six real-world wayfinding routes tested, increasing in difficulty from route 1 to route 6. (E) Two participant viewpoints from within Sea Hero Quest. On the left is a completely unobscured level; on the right is a completely obscured (fog) level. (F) Example participant viewpoint during route 1 of the real-world wayfinding task, the streets visible here are Museum Street and Great Russell Street. (G) Example trajectories taken by two different participants on level 11 of Sea Hero Quest and route 4 in the real-world task.

in order to aid older participants learn the game controls, we decided to present participants with screenshots of the controls, while performing the associated action (i.e., tapping or swiping). To make sure that our sample was representative of the wider population on video game familiarity/ visuo-motor skill, we compared the performance on Level 1 to the distribution of performance on Level 1 for the world sample of 3.8 million players.

Participants completed six levels of Sea Hero Quest (1, 6, 8, 11, 16, and 43). These levels were previously chosen when comparing virtual and real-world navigation in a younger cohort on the basis of varying difficulty [30,44]. We included Level 1 as a training level, and Level 43 as a high difficulty level. Level 8 was not included in the previous study with the younger cohort, and has thus been excluded from the analysis.

Performance on wayfinding levels was quantified by calculating the Euclidean distance, point-to-point, travelled in each level (in virtual metres). These points were assessed from participants' trajectories, coordinates of which were sampled at Fs = 2 Hz. To measure total

gameplay distance travelled, we summed the Euclidean distances from levels 6 to 43 (excluding Level 1, which functioned as a training level and as a measure of game-specific usability). In a separate analysis, we used Level 1 to compare the performance of our sample to the age-matched sample from the global dataset.

### Real-world task

To control for any order effect, the task order is counterbalanced across participants, with half of participants completing the virtual task followed by the real-world task and the other half completing the real-world task followed by the virtual task. The real-world wayfinding task consisted of 6 trials of increasing difficulty contained within the Covent Garden area of London (see Fig 1B,E). Difficulty was determined via a combination of the following variables: number of streets, number of goals, street network complexity, and goal arrangement complexity (following the experimental design in [30]). At the start of each wayfinding trial, participants were led to distinct starting locations corresponding to each trial. To control for participants walking down a street contained in one of the trials before being tested on that route, we led participants around the testing area to each starting location. Testing primarily occurred on side and back streets to avoid high traffic areas and potential familiarity from salient landmarks. No participants indicated high familiarity with the area tested. Once each starting location was reached, participants were directed to face in a specific direction and were then shown a map that indicated their starting location (and facing direction), a simplified network of the streets contained in the trial, and the location and order (indicated by number) of the goals (numbered doorways). Maps were displayed on a laminated A4 sheet and were held by the experimenter to prevent participants turning the map. Participants were instructed that after a maximum study time of 60 seconds, they would receive colour photographs of doors that correspond to each of the numbered goals -- which they would be able to hold on to for the duration of the trial. After 60 seconds, or when the participant finished studying the map, the map was removed, and the time limit began to locate each of the goals. During piloting, we checked if participants would be able to complete each trial under the same time restrictions as the one used in the previous protocol [30]. Confirming that this was possible while still allowing for a few errors, we kept the time restrictions the same for each trial (Trial 1: 6 minutes, Trial 2: 6 minutes, Trial 3: 6:30 minutes, Trial 4: 6:30 minutes, Trial 5: 12 minutes, Trial 6: 14 minutes). Participants were instructed that if they reached a blurred-out area on the route map, the experimenter would indicate the edge of the search area had been reached and they must turn around. If the participant did not locate all goals before the time limit, they were instructed that the trial had ended and then directed to the starting location of the next trial.

For each trial, we measured distance travelled using Q-Starz GPS travel recorders sampling at Fs = 5 Hz. All our performance data consisted of trajectory data (e.g., there was no data about real-time decision-making). For subsequent analysis, we considered both total distance travelled and normalised distance, as defined by distance divided by the proportion of goals reached. Performance was quantified with point-to-point sums of the Euclidean distance travelled for each trial (in metres).

### Statistical analysis

To test whether Sea Hero Quest is predictive of real-world navigation ability, we calculated Pearson's correlations between real-world wayfinding performance and the virtual performance at each SHQ wayfinding level.

To assess the cognitive decline in the real and the virtual environments, we compared the performance of older versus younger participants. In both environments, we used a chi-square

test to compare the proportions of participants in the older versus younger cohorts who completed the task before the time limit.

We also examined whether familiarity with the video game controls might play a role in driving navigation performance. To explore this, we examined whether the training level (Level 1) was correlated with real-world performance for our older group. Since Level 1 is not an assessment of spatial navigation ability but rather video game familiarity/ visuo-motor skill (the goal remains visible from the start of the Level), we used Pearson's correlation analysis to test whether there was a correlation between the distance travelled in Level 1 and the normalised distance in the real world.

## Results

The difficulty of the Sea Hero Quest levels and real-world wayfinding routes for a younger [30] and the current sample are reported in Table 1. Difficulty in the Sea Hero Quest levels is reported as the percentage of participants able to complete the level before the helping arrow appeared. Difficulty in the real-world wayfinding task is reported as the percentage of goals reached by participants before the time limit. For Sea Hero Quest, this percentage decreases from 100% on the first level to 47% on level 43 for the younger sample; and for the older sample, this percentage decreases from 100% on level 1 to 5% on level 43 (only 1 participant was able to complete the final level before the arrow appeared). For the real-world wayfinding task, the percentage of goals reached decreases from 99% on route 1 to 74% on route 6 for the younger sample; this percentage decreases from 100% on route 1 to 78.8% on route 6 for the older sample.

### Correlation between virtual wayfinding and real-world wayfinding

We calculated Pearson's correlations between real-world wayfinding performance and the virtual performance at each SHQ wayfinding level (see Fig 2). We aggregate real-world

**Table 1. SHQ performance vs real-world task performance.** Column 1 (Level) indicates the level in SHQ. Column 2 (Y) indicates the proportion of participants in the younger cohort who completed the virtual task before the time limit. Column 3 (O) indicates the proportion of participants in the older cohort who completed the virtual task before the time limit. Column 4 (Y vs O) is the result of a chi-square test to compare the proportions of participants in the older vs younger cohorts who completed the virtual task before the time limit. For level 1, the value is undefined because it is the same number in both cohorts. Column 5 (Route) indicates the route number in the real-world task. Column 6 (Y) indicates the proportion of participants in the younger cohort who completed the real-world task before the time limit. Column 7 (O) indicates the proportion of participants in the older cohort who completed the real-world task before the time limit. Column 8 (Y vs O) indicates the result of a chi-square test to compare the proportions of participants in the older vs younger cohorts who completed the real-world task before the time limit.

| Level | Y | O | Y vs O | Route | Y | O | Y vs O |
|-------|------|------|-------------------------|-------|------|------|-------------------------|
| 1 | 1.0 | 1.0 | | 1 | 0.99 | 1.0 | p = 0.29 chi = 1.09 |
| 6 | 0.97 | 0.65 | p = 3.8e-4 chi = 12.59 | 2 | 0.93 | 1.0 | p = 0.52 chi = 0.40 |
| 11 | 0.67 | 0.10 | p = 3.13e-5 chi = 17.33 | 3 | 0.92 | 0.98 | p = 0.67 chi = 0.17 |
| 16 | 0.90 | 0.40 | p = 1.5e-5 chi = 18.73 | 4 | 0.91 | 0.83 | p = 0.56 chi = 0.33 |
| 43 | 0.47 | 0.05 | p = 0.002 chi = 9.67 | 5 | 0.88 | 0.89 | p = 0.78 chi = 0.078 |
| | | | | 6 | 0.74 | 0.79 | p = 0.88 chi = 0.02 |

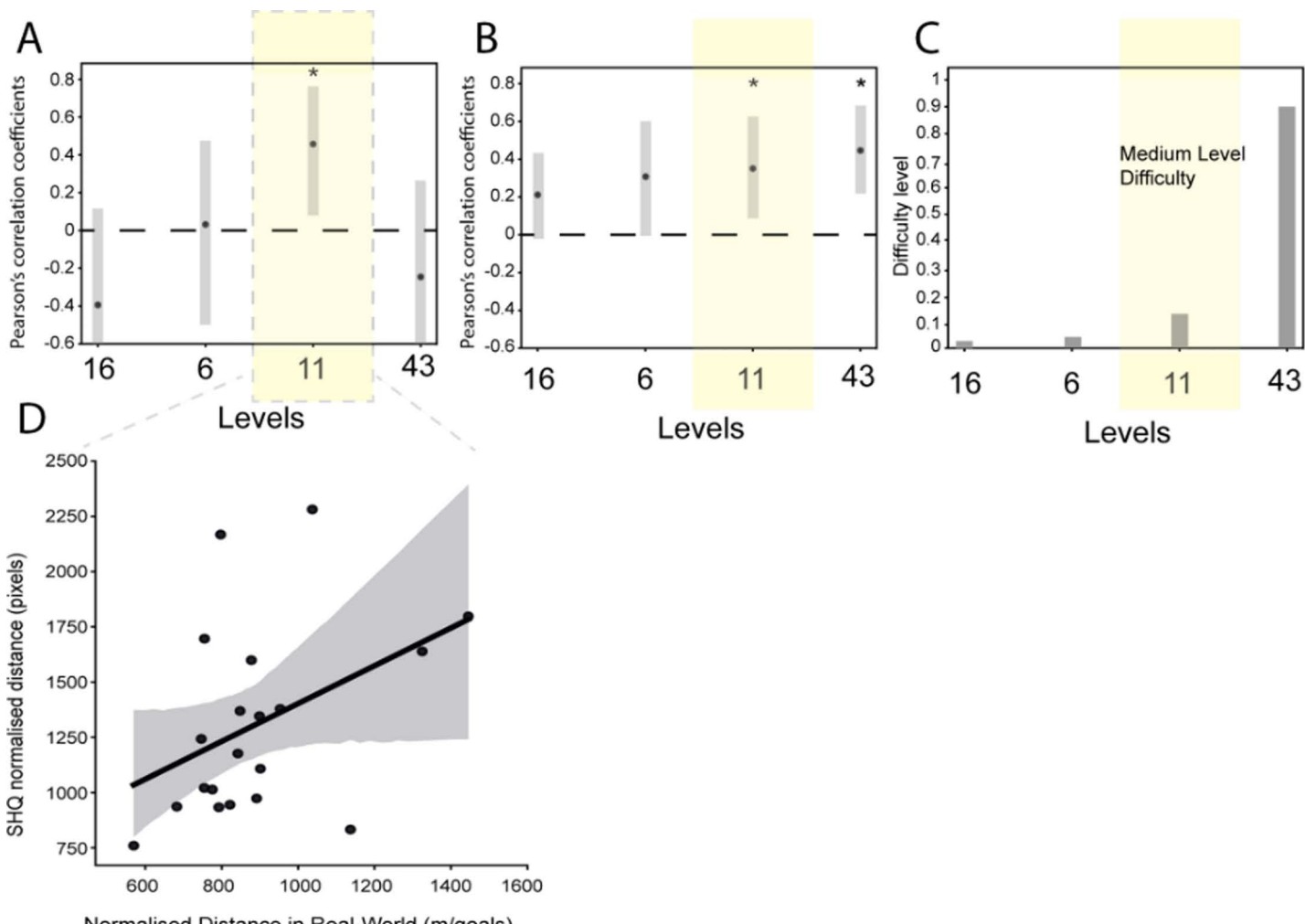

**Fig 2. SHQ predicts real-world wayfinding performance in older participants for medium difficulty levels.** (A) Correlation coefficient and 95% confidence intervals for older participants for each level of Sea Hero Quest compared with real-world navigation (normalised metres travelled). The amber area indicates the medium difficulty level (11). (B) Correlation coefficients and 95% confidence intervals for younger participants (based on [30]) for each level of Sea Hero Quest compared with real-world navigation (normalised metres travelled). The amber area indicates the medium difficulty level (11). (C) Level difficulty of each SHQ level (based on [44]). The amber area indicates the medium difficulty levels (11). For each level, difficulty is defined by subtracting the minimum trajectory length from the median trajectory length and then normalising it with the minimum trajectory length [44]. The obtained difficulty values are then scaled between 0 and 1. Note that the "level difficulty" metric is not the same as the "difficulty" metric used to assess the performance of each participant. (D) Correlation coefficient for the normalised distance in Level 11 of Sea Hero Quest compared with real-world navigation.

performance as the variation in environmental options/goals was less variable than in Sea Hero Quest, dictated by the fact we were using London streets. Past work has also emphasised differences with different SHQ levels (e.g., Lim et al., 2023). We found the correlation was not significant for the easiest levels (level 16: r = -0.20, 95%CI = [−0.43, 0.32], p = 0.76; level 6: r = −0.08, 95%CI = [−0.33, 0.21], p = 0.66), or for the hardest level (level 43: r = −0.06, 95% CI = [−0.38, 0.13], p = 0.89), but was significant for level 11 (r = 0.41, 95%CI = [−0.06, 0.75], p = 0.03), which was a medium difficulty level. No outliers were identified.

A Pearson's correlation analysis confirmed the non-significant relationship between level 1 and real-world performance (r = -0.06, 95%CI = [−0.55 0.31], p = 0.88). Thus, being good at learning the game controls had no relationship to real-world wayfinding.

Participants also completed the NSQ. This questionnaire asks participants to self-rate the strategies they use for navigating, such as a focus on maps vs a focus on routes. We found a negative correlation between NSQ scores and distance in Sea Hero Quest (Pearson's r = −0.63, p = 0.02), which indicates that participants with higher scores (indicating a tendency towards map-based navigation strategies) perform better in the virtual task. There was no significant correlation between NSQ scores and real-world performance (r = −0.11, p = 0.71). Anecdotally, some older participants mentioned looking to see if the goal (doorway) was even or odd numbered and then focusing on the side of the road to focus the search to be more efficient. This is because, in the UK, one side of the road will be even numbers and the other side odd numbers. It is possible this was also a strategy used by some younger participants but was not documented.

## Comparison to the population-level dataset (286,828 matched participants)

To assess whether our sample was representative of the wider population on visuo-motor skill/video game familiarity, we compared the duration (seconds) of completion on Level 1 to the distribution of duration on Level 1 for the world sample of 3.8 million players [8]. Since we tested an older population, we compared performance only for the population within the age range of 50 to 75 years old. The full Sea Hero Quest data set sample is 286,828 after filtering the age range. We found no difference between our sample trial duration and that of the matched age population from the mass global dataset, according to Welch's t-test (t = 1.275, p = 0.217). Thus, our sample of 20 participants was consistent with the broader population in terms of capacity to learn level 1.

## Discussion

We explored whether an app-based virtual navigation test, Sea Hero Quest, would be predictive of real-world navigation in an older population (50-75 yrs). We compared navigation performance in Sea Hero Quest with performance in an analogous task in the Covent Garden area of London, UK. We found that some, but not all, Sea Hero Quest levels were able to predict real-world wayfinding performance in our older population.

Our key finding was that when dividing the levels into easy, medium and difficult, only the medium difficulty level was able to predict wayfinding performance in the streets of London, UK. One possibility we considered was that older participants might need more exposure to the SHQ task to get familiar with the task, due to less experience with virtual environments, and thus the correlation with the medium difficulty level occurs because participants experience it later in the sequence. However, this explanation seems less likely, as the easiest level (level 16) was experienced later in testing than the medium level (level 11). Rather, the success of the medium level may be due to a near ceiling effect for easy levels, and, on the other hand, a near floor effect for the hardest level. This stands in contrast to the younger cohort in a previous study, for which there was only a mild ceiling effect and no floor effect [30]. The emerging picture is that of a 'Goldilocks' effect in predictability for older participants. There is a sweet spot of middle difficulty when it comes to the predictability of virtual navigation tests for an older population. Overall, our result is broadly consistent with existing studies showing navigation tests in both desktop and immersive VR can be predictive of real-world navigation performance [11,13,15,17,30,45–48].

Evidence that a mid-difficulty Sea Hero Quest environment is predictive of real-world wayfinding for both older and younger participants may be useful in predicting errors in the real-world for pre-clinical/ clinical populations, such as those at genetic risk of Alzheimer's disease [22,38], diagnosed Alzheimer's disease [17,22], amnesic cases [49] and traumatic brain injury

cases [40]. In these prior studies many of the participants are older than the young cohort tested in [30]. In future research, it may be useful to develop more varied real-world tasks that require detours and shortcuts, leading to greater variation in navigation performance and separate cognitive capacities [50–53].

Our older population varied in how much they preferred using maps or routes for navigating, as identified using the Navigation Strategy Questionnaire [42]. While we recently found no correlation between the extent to which participants preferred maps to navigate and Sea Hero Quest performance [21], we did find this for our older group. It is possible this correlation emerges in older participants due to the increased shift towards egocentric strategies in the older population [54–62] (for a review, see [63]), leading to greater variation in strategies and behaviour in the old population.

While we found that, as expected, the younger cohort outperformed the older cohort in the virtual task, this was not the case for the real-world task, where performance was similar. This finding conflicts with the currently prevailing understanding of a decline in navigation performance with age [7,31–36,62,64]. This may be in part due to self-selection bias within our sample. Older adults with higher cognitive function may be more predisposed to participate in a psychological study. Moreover, all participants needed to be able to walk unaided for up to two and a half hours, biasing our sample to active, healthier older adults. This explanation is supported by the mean score of 28.45 in the Montreal Cognitive Assessment in our population, above what would be expected for a population this age (26.3, [65]). Note that self-selection bias – not necessarily of physically active older participants, but rather of older participants with a higher-than-average cognitive function – might also affect the global sample of Sea Hero Quest players, where there is higher variability after the age of 60 [7] (for an exploration of selection bias in this age group, see [25]). Our results mirror those of Hill and colleagues [66], who found older participants' decline in navigation ability was mitigated when the virtual world was presented in a more naturalistic medium (immersive VR) compared to when it was presented on a flat desk-top screen.

Selection bias may partly explain why the younger cohort does not outperform the older cohort in the real-world wayfinding task. However, when it comes to the correlation between virtual and real-world navigation performance for this older cohort, the picture is more complicated. Put simply, why are these highly active, older participants with high functioning cognitive capacities struggling with the harder level of Sea Hero Quest, but not with navigating the streets of London? A tempting explanation is that older participants struggle with video games, but note that we used Level 1 (which is not an assessment of navigation ability) to assess baseline videogame skill, and, as expected, we did not find a relationship between the distance travelled in Level 1 and the normalised distance in the real world. We also ensured that participants were not highly familiar with the area tested in Covent Garden, so familiarity alone is unlikely to explain the difference. Notably, the streets used were not those that would be familiar to a regular visitor to Covent Garden. Rather they were backstreets between the British Museum and the main Covent Garden area [30]. A possible explanation is that older participants develop compensatory navigational strategies, effective in the real-world, but that do not transfer as well to an unfamiliar virtual environment. For example, the anecdotal reports by several older participants of using the doorway numbers (even vs odd) to focus search on one side of the street give an example of a useful strategy. Older participants may have also developed better predictions (that may not be conscious) for what to expect in a London street network, a more accurate spatial schema of London [67]. Finally, it is likely that our older participant group will have gained more experience with viewing a paper map and then navigating from memory compared to the younger cohort tested in [30]. More research to explore how schemas and strategies feature in real-world wayfinding alongside

more computational approaches to spatial behaviour would be beneficial to better understand adaptations to support navigation [68–70].

Our results align with some early wayfinding studies of older populations. Younger participants outperformed older participants in lab-based measures of spatial ability, but not in a realistic wayfinding task in a hospital setting [71]. In another study, younger subjects outperformed older subjects in perspective-taking and distance-estimation tasks, but performed at equivalent levels when the tests presented an array of stimuli that resembled the subjects' hometowns [72]. This line of studies supports the idea that older participants might develop specific navigation strategies adapted to their local environment that are hard to capture with traditional tests [73]. Another study of older deer hunters in Nova Scotia found that they were in fact less likely to get lost than younger deer hunters [74]. The elderly hunters who became lost covered similar distances and were just as likely to reorient themselves as were younger hunters, which the researchers attributed to compensatory navigation strategies. In traditional cultures navigation skill is typically entwined with knowledge of the environment such that navigation skill improves with age [75]. In more recent studies of western participants, older expert orienteers showed compensatory strategies of attention that outweighed cognitive decline, so that they outperformed non-expert older subjects in an attention task [76], and older expert drivers developed compensatory strategies that mitigated possible collision risk, when compared with non-expert drivers both mid-aged and old [77]. This echoes the role of compensatory strategies in expert performance in other domains: older chess players compensate memory deficits by considering fewer alternative moves but in a more efficient way [78], and older typists planned ahead more efficiently by actively scanning the text to be typed as they worked [79].

In conclusion, we show that, for older participants, when the virtual navigation is not too taxing, but not too trivial, it can provide a good prediction of real-world performance. We also find older participants perform well against the younger participants when tested in the real-world, but much worse when tested in the virtual environment. Two limitations of our study is the likely self-selection bias in our cohort of older participants, and the fact that we did not design the experiment to study compensatory skills in the older participants. It is likely that our group of older participants had compensatory skills they could deploy in the real-world task but that did not translate to the hardest level in our virtual task. This factor should be considered hand in hand with the likely self-selection bias of our sample (our participants being more physically active and having higher cognitive capacities than average). The two factors are, in fact, complementary. The self-selection bias can explain the stronger than expected performance of our older cohort at the real-world task, and much of this strong performance is likely to be due to compensatory skills and strategies that do not transfer well to the more difficult levels of Sea Hero Quest. A further limitation is that the sample size is relatively small, because of the difficulty associated with recruiting older participants for this lengthy and physical task. Future work should try to overcome the limitations of selection bias and explore more in detail the nature and role of compensatory strategies in navigational tasks for older participants. For instance, studies could employ cognitive ethnographic methods to identify compensatory strategies, and then design psychology studies to test the impact of those strategies in navigation performance (for a recent example of cognitive ethnography being employed to study compensatory strategies, see [80]). Future studies should also explore in more detail the influence of task difficulty on real-world validity for older populations in varied navigation tasks.

## Acknowledgements

We would like to thank all the participants who volunteered to take part in this research. We thank Jane Cooksey for help with recruitment.

## Author contributions

**Conceptualization:** Sarah Goodroe, Jan Wiener, Antoine Coutrot, Michael Hornberger, Hugo J Spiers.

**Data curation:** Sarah Goodroe, Christoffer J Gahnstrom, Antoine Coutrot.

**Formal analysis:** Sarah Goodroe, Christoffer J Gahnstrom, Antoine Coutrot.

**Funding acquisition:** Michael Hornberger, Hugo J Spiers.

**Investigation:** Sarah Goodroe, Christoffer J Gahnstrom, Antoine Coutrot, Michael Hornberger, Hugo J Spiers.

**Methodology:** Sarah Goodroe, Christoffer J Gahnstrom, Jan Wiener, Antoine Coutrot, Hugo J Spiers.

**Project administration:** Pablo Fernandez Velasco.

**Supervision:** Hugo J Spiers.

**Visualization:** Sarah Goodroe, Pablo Fernandez Velasco, Antoine Coutrot, Hugo J Spiers.

**Writing – original draft:** Sarah Goodroe, Pablo Fernandez Velasco.

**Writing – review & editing:** Pablo Fernandez Velasco, Christoffer J Gahnstrom, Jan Wiener, Antoine Coutrot, Michael Hornberger, Hugo J Spiers.

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
