## [Decision Letter · Decision Letter 0]

25 Oct 2024

PONE-D-24-38652Predicting real-world navigation performance from a virtual navigation task in older adultsPLOS ONE

Dear Dr. Fernandez Velasco,

Thank you for submitting your manuscript to PLOS ONE. After careful consideration, we feel that it has merit but does not fully meet PLOS ONE’s publication criteria as it currently stands. Therefore, we invite you to submit a revised version of the manuscript that addresses the points raised during the review process.

The paper addresses a timely and relevant topic. The research problem, objectives, methodology, and overall discussion are of sufficient quality to warrant moving the manuscript forward to a major revision.

All reviewer suggestions should be addressed before the paper can be accepted. In particular, greater clarity is needed in articulating the study's goals and hypotheses, specifically how the research addresses whether Sea Hero Quest can predict real-world navigation ability in an older population.

The sample size in the study is quite small, and while the authors acknowledge this limitation at the end of the paper, they should exercise more caution in reporting their findings. With such a small sample, the authors can only identify behavioral tendencies, and this should be emphasized more clearly throughout the manuscript.

Additionally, please include a reference for the Navigational Strategy Questionnaire (NSQ).

We look forward to receiving your revised manuscript.

Kind regards,

Sara Eloy, Ph.D

Academic Editor

PLOS ONE

Journal Requirements:

2. Thank you for stating the following financial disclosure: This research is part of the Sea Hero Quest initiative funded and supported by Deutsche Telekom. Alzheimer’s Research UK (ARUK-DT2016-1) funded support for the research; Glitchers designed and produced the game; and Saatchi and Saatchi London managed its creation.  

3. Thank you for stating the following in the Acknowledgments Section of your manuscript: We would like to thank all the participants who volunteered to take part in this research. This research is part of the Sea Hero Quest initiative funded and supported by Deutsche Telekom. Alzheimer’s Research UK (ARUK-DT2016-1) funded support for the research; Glitchers designed and produced the game; and Saatchi and Saatchi London managed its creation. We thank Jane Cooksey for help with recruitment.

Please remove any funding-related text from the manuscript and let us know how you would like to update your Funding Statement. Currently, your Funding Statement reads as follows: This research is part of the Sea Hero Quest initiative funded and supported by Deutsche Telekom. Alzheimer’s Research UK (ARUK-DT2016-1) funded support for the research; Glitchers designed and produced the game; and Saatchi and Saatchi London managed its creation. 

4. Thank you for uploading your study's underlying data set. Unfortunately, the repository you have noted in your Data Availability statement does not qualify as an acceptable data repository according to PLOS's standards.

Reviewers' comments:

Reviewer's Responses to Questions

**Comments to the Author**

1. Is the manuscript technically sound, and do the data support the conclusions?

Reviewer #1: Yes

Reviewer #2: Yes

2. Has the statistical analysis been performed appropriately and rigorously? 

Reviewer #1: Yes

Reviewer #2: Yes

3. Have the authors made all data underlying the findings in their manuscript fully available?

Reviewer #1: Yes

Reviewer #2: Yes

4. Is the manuscript presented in an intelligible fashion and written in standard English?

Reviewer #1: Yes

Reviewer #2: Yes

5. Review Comments to the Author

Reviewer #1: Review Comments to the Author:

This study explores the link between virtual navigation skills and real-world wayfinding abilities in older adults. The research aims to understand cognitive navigation in different environments, providing insights into aging and potential strategies to reduce cognitive decline. By using virtual and real-world tasks, the study shows how performance in a controlled setting may reflect actual navigation abilities, contributing to the development of interventions for improving spatial awareness and independence in older populations. It also highlights the potential of virtual reality as a tool for assessing cognitive function and sets the stage for future research using virtual reality to address challenges faced by older adults.

Although the manuscript is clear, it would benefit from some minor revisions.

1. Technical Soundness of the Manuscript:

The manuscript provides a thorough study of cognitive navigation across different age groups, comparing performance in real-world and virtual tasks. The methodologies are generally appropriate, and including both real-world and virtual navigation tasks adds depth to the study.

However, it is unclear whether the researcher can monitor the participants' decisions in real-time during the virtual pathfinding tasks. While the game collects data on participants' movements, decisions, and actions during the navigation tasks, the manuscript does not explicitly explain how this data is collected or whether it includes real-time decision-making. Clarifying this aspect would enhance the understanding of how the virtual wayfinding data is gathered and interpreted

2. Data Availability:

The authors have provided a link to the data, which supports transparency. Despite not providing a description in the manuscript the online data available provides a brief description of the dataset structure in a supplementary document. This helps readers who are not familiar with the data to navigate and understand it more effectively.

3. Presentation and Language:

The manuscript is well-organized, but certain sections could be presented more clearly, especially when discussing complex results.

Suggestions for Improvement:

•Abstract: The abstract is well-written, but simplifying it would improve its accessibility for readers unfamiliar with navigation studies. Shortening some of the background information in the abstract could also make it more focused.

•Introduction: While the introduction establishes a strong foundation, it includes some repetition. Streamlining this section and clearly outlining the research question early on would enhance the focus.

•Methods: Althout part of previous papers and research, please provide additional details about the experimental setup, specifically regarding the virtual task environment. It would be helpful to explain the interface used in the Sea Hero Quest game in order to better clarify how participants interacted with the virtual environment.

4. Discussion:

The study's main finding is that medium-difficulty tasks in a virtual environment can predict real-world navigation in older adults. The discussion introduces the Goldilocks effect, describing how tasks of moderate difficulty hit the "sweet spot" for predictability. The authors effectively connect their findings to previous research, situating their results within the broader context of aging, navigation, and compensatory strategies. This provides a clear foundation for understanding how older adults may perform in both virtual and real-world environments. Additionally, the discussion considers compensatory strategies that older participants may use in real-world navigation but find difficult to apply in virtual settings, highlighting how familiarity with certain environments can mask declines in other areas. By addressing the issue of selection bias, where healthier and more cognitively active older adults may have participated, the authors reinforce the credibility of their conclusions. The recommendations for future research offer valuable insight for the next steps in this field of study, such as exploring more complex navigation tasks and examining cognitive capacities such as spatial strategies and schemas.

The discussion effectively interprets the data but would benefit from a more structured approach and further exploration of the implications of the findings.

Suggestions for Improvement:

• Clarify Key Findings: The discussion would benefit from a brief summary of the main findings, followed by a clear differentiation between primary results and secondary interpretations. This will aid in comprehending the fundamental contributions of the research.

•Limitations and Future Directions: The limitations section is concise and could benefit from further expansion.

5. Charts and Tables:

The charts are useful, althowg not simple to understand they are clearly explained in the legends, effectively supporting the results.

6. References:

The references are comprehensive and relevant to the research. However, some minor improvements could be made.

Suggestions for Improvement:

•In-text Citations: Ensure that all studies mentioned in the discussion are appropriately cited in the references section. There were a few instances where prior work was mentioned without direct references.

•Reference Formatting: Double-check the consistency of the reference formatting (e.g., author names, journal titles, and volume/issue numbers) to ensure they meet the journal’s style guidelines.

Reviewer #2: A very interesting study combining navigation testing in a virtual environment versus a real-world navigation task in older adults. The introduction and scientific background are sound and the authors are familiar with relevant literature. I want to note that I am very excited about this research and believe in the value of this topic. Furthermore, open science practices via OSF is applauded. However, regarding statistics and combining the aims and the results, I have some comments.

General comments:

• I notice variations in the terminology used: spatial ability, spatial cognition, navigation, spatial navigation, wayfinding… Is there a way to make this more coherent, or to describe how spatial cognition and navigation are related to each other?

• When reading the title, I expect to see an analysis on how navigation performance by means of the Sea Hero Quest could predict real-world navigation performance. However, when looking at the text, this is not mentioned. The closest thing appears to be Figure 2, Panel D. I think the study is good and the title is interesting as well, but I am missing the link between both. Please elaborate.

Specific comments:

Introduction

• “This is contrast to studies using object-based spatial tasks, which have mixed results in predicting real-world spatial behaviour (e.g. path integration, wayfinding)”: To make it more clear, can you add an example of object-based spatial tasks?

• “Whilst this is important to show ecological validity of an app-based assessment, the extent to which it could predict real-world navigation in later life is unclear due to the difference in navigation ability that occurs with age”: Can you make this more concrete by explaining what difference you expect in navigation ability in older age?

• “Furthermore, a significant value in developing tests of navigation is that they may be useful in the early diagnosis and cognitive monitoring of Alzhiemers disease or other neurological conditions”: Please correct the typo in Alzheimer's disease. Please also shortly describe why navigation is interesting in Alzheimer's disease, for example by focusing on its specificity.

• Aim of the study: I think the aim is clear and very interesting. I would like to see the hypothesis more concrete by describing how this decline in navigation performance would be conceptualized. Will you focus on accuracy for example or will you focus on the navigation strategy applied (measuring a shift from allo- to egocentric navigation)? This will become more clear in the rest of the paper but it would be nice to have this as concrete as possible in the introduction already.

Methods – statistical analysis: I miss how you will analyze your research question, namely whether Sea Hero Quest is predictive of real-world navigation ability in an older population. As you are also expecting to find a general decline reflected in a difference between the older group and the younger group of your previous study, this also has to be described in the statistical analysis part how you will test this. If you are including testing for differences on the basis of gender, this also has to be included in the aims at the end of the introduction.

Results:

• Table 1: Please describe what each column represents.

• Paragraph “Comparison to the population-level dataset”: If you want to analyze this, please also add this as an aim of this study in the introduction, as well as in the statistical analysis.

Discussion:

“While we found that, as expected, the younger cohort outperformed the older cohort in the virtual task, this was not the case for the real-world task, where performance was similar.”: Please correct me if I’m wrong, but where do I find the analysis in the results section? Table 1 comes close but I am missing a statistical analysis on these data.

“Future work should (…) and explore more in detail the nature and role of compensatory strategies (…)”: Can you make this more concrete? How would you operationalize this?

6. PLOS authors have the option to publish the peer review history of their article (what does this mean? ). If published, this will include your full peer review and any attached files.

**Do you want your identity to be public for this peer review?** For information about this choice, including consent withdrawal, please see our Privacy Policy .

Reviewer #1: No

Reviewer #2: **Yes: ** Joyce Bosmans

---

## [Author Response · Author response to Decision Letter 0]

11 Nov 2024

We are grateful to the reviewers for their insightful comments. We have revised the manuscript in light of these recommendations, and we believe the resulting manuscript is in a much stronger position. Below, we outline how we have addressed the comments by the reviewers. In the attached "Response to Reviewers" document, we include the original review comments in black font, our reply in blue font, and quotes from our manuscript in red font. Overall, we have made the following revisions:

Reviewer #1: Review Comments to the Author:

This study explores the link between virtual navigation skills and real-world wayfinding abilities in older adults. The research aims to understand cognitive navigation in different environments, providing insights into ageing and potential strategies to reduce cognitive decline. By using virtual and real-world tasks, the study shows how performance in a controlled setting may reflect actual navigation abilities, contributing to the development of interventions for improving spatial awareness and independence in older populations. It also highlights the potential of virtual reality as a tool for assessing cognitive function and sets the stage for future research using virtual reality to address challenges faced by older adults.

We are grateful for this positive assessment of our manuscript.

Although the manuscript is clear, it would benefit from some minor revisions.

1. Technical Soundness of the Manuscript:

The manuscript provides a thorough study of cognitive navigation across different age groups, comparing performance in real-world and virtual tasks. The methodologies are generally appropriate, and including both real-world and virtual navigation tasks adds depth to the study.

However, it is unclear whether the researcher can monitor the participants' decisions in real-time during the virtual pathfinding tasks. While the game collects data on participants' movements, decisions, and actions during the navigation tasks, the manuscript does not explicitly explain how this data is collected or whether it includes real-time decision-making. Clarifying this aspect would enhance the understanding of how the virtual wayfinding data is gathered and interpreted.

We have edited the part of our manuscript in which we explain the data we collected from participants. It now reads as follows:

For each trial, we measured distance travelled using Q-Starz GPS travel recorders sampling at Fs = 5 Hz. All our performance data consisted of trajectory data (e.g. there was no data about real-time decision-making). For subsequent analysis, we considered both total distance travelled and normalised distance, as defined by distance divided by the proportion of goals reached. Performance was quantified with point-to-point sums of the Euclidean distance travelled for each trial (in metres).

2. Data Availability:

The authors have provided a link to the data, which supports transparency. Despite not providing a description in the manuscript the online data available provides a brief description of the dataset structure in a supplementary document. This helps readers who are not familiar with the data to navigate and understand it more effectively.

3. Presentation and Language:

The manuscript is well-organized, but certain sections could be presented more clearly, especially when discussing complex results.

Suggestions for Improvement:

•Abstract: The abstract is well-written, but simplifying it would improve its accessibility for readers unfamiliar with navigation studies. Shortening some of the background information in the abstract could also make it more focused.

We have simplified and shortened the abstract:

Virtual reality environments presented on tablets and smartphones offer a novel way of measuring navigation skill and predicting real-world navigation problems. The extent to which such virtual tests are effective at predicting real-world navigation in older populations remains unclear. We compared the performance of 20 older participants (54-74 years old) in wayfinding tasks in a real-world environment in London, UK, and in similar tasks designed in a mobile app-based test of navigation (Sea Hero Quest). In a previous study with young participants (18-35 years old), we were able to predict navigation performance in real-world tasks in London and Paris using this mobile app. We find that for the older cohort, virtual navigation performance predicts real-world performance for medium difficulty, but not for the easy or difficult environments. Overall, our study supports the utility of using digital tests of spatial cognition in older age groups, while carefully adapting the task difficulty to the population.

•Introduction: While the introduction establishes a strong foundation, it includes some repetition. Streamlining this section and clearly outlining the research question early on would enhance the focus.

We have restructured and streamlined the introduction. We now discuss the promise and challenges of using virtual environments in empirical psychology first, and then we move onto a shortened description of SHQ and of existing work using the app.

•Methods: Although part of previous papers and research, please provide additional details about the experimental setup, specifically regarding the virtual task environment. It would be helpful to explain the interface used in the Sea Hero Quest game in order to better clarify how participants interacted with the virtual environment.

We have rewritten the methods to clarify the experimental setup. Specifically, this is the new opening paragraph describing the virtual task:

We designed a mobile video game, Sea Hero Quest, to measure spatial navigation ability. Participants navigate a boat in a virtual environment, using either an Acer tablet or an iPad as an interface (for an extensive description, see Coutrot et al., 2018). At the start of each level, participants viewed a map displaying their current position and ordered goal locations in navigable waterways of the environment. In this task, participants were able to study the map without any time restriction. When done studying, the participants pressed play and then (from a view that positionied the player's perspective behind the boat) had to navigate to the goal locations in the order indicated on the map by number (i.e. find goal 1, find goal 2, … find goal n, until all n goals have been located), see Figure 1A,E. In the virtual water network, goals corresponded to buoys with flags signalling the goal number, with a circle around the buoy indicating the goal area. Participants had to enter the corresponding circle to successfully reach each goal. Once all goals were reached, the level was registered as complete, and participants moved on to the next level, until all six levels were completed. For each level, a guidance arrow appears on the screen after a certain duration, indicating the direction to the goal along the vector to the goal. This arrow appeared after the following durations by level: 80s in Level 1, 70s in Level 6, 80s in Level 11, 80s in Level 16, and 200s in Level 43.

4. Discussion:

The study's main finding is that medium-difficulty tasks in a virtual environment can predict real-world navigation in older adults. The discussion introduces the Goldilocks effect, describing how tasks of moderate difficulty hit the "sweet spot" for predictability. The authors effectively connect their findings to previous research, situating their results within the broader context of aging, navigation, and compensatory strategies. This provides a clear foundation for understanding how older adults may perform in both virtual and real-world environments. Additionally, the discussion considers compensatory strategies that older participants may use in real-world navigation but find difficult to apply in virtual settings, highlighting how familiarity with certain environments can mask declines in other areas. By addressing the issue of selection bias, where healthier and more cognitively active older adults may have participated, the authors reinforce the credibility of their conclusions. The recommendations for future research offer valuable insight for the next steps in this field of study, such as exploring more complex navigation tasks and examining cognitive capacities such as spatial strategies and schemas.

Thank you for highlighting this in the review process.

The discussion effectively interprets the data but would benefit from a more structured approach and further exploration of the implications of the findings.

Suggestions for Improvement:

• Clarify Key Findings: The discussion would benefit from a brief summary of the main findings, followed by a clear differentiation between primary results and secondary interpretations. This will aid in comprehending the fundamental contributions of the research.

We have rewritten the start of the discussion section to make it clear that our key finding is that the medium level in the virtual task, but not the easy or difficult ones, were able to predict wayfinding performance in the real-world task:

We explored whether an app-based virtual navigation test, Sea Hero Quest, would be predictive of real-world navigation in an older population (50-75 yrs). We compared navigation performance in Sea Hero Quest with performance in an analogous task in the Covent Garden area of London, UK. We found that some, but not all, Sea Hero Quest levels were able to predict real-world wayfinding performance in our older population.

Our key finding was that when dividing the levels into easy, medium and difficult, only the medium difficulty level was able to predict wayfinding performance in the streets of London, UK.

•Limitations and Future Directions: The limitations section is concise and could benefit from further expansion.

We have expanded the limitations section considerably:

In conclusion, we show that, for older participants, when the virtual navigation is not too taxing, but not too trivial, it can provide a good prediction of real-world performance. We also find older participants perform well against the younger participants when tested in the real-world, but much worse when tested in the virtual environment. Two limitations of our study is the likely self-selection bias in our cohort of older participants, and the fact that we did not design the experiment to study compensatory skills in the older participants. It is likely that our group of older participants had compensatory skills they could deploy in the real-world task but that did not translate to the hardest level in our virtual task. This factor should be considered hand in hand with the likely self-selection bias of our sample (our participants likely being more physically active and having higher cognitive capacities than average). The two factors are, in fact, complementary. The self-selection bias can explain the stronger than expected performance of our older cohort at the real-world task, and much of this strong performance is likely to be due to compensatory skills and strategies that do not transfer well to the more difficult levels of Sea Hero Quest. A further limitation is that the sample size is relatively small, because of the difficulty associated with recruiting older participants for this lengthy and physical task. Future work should try to overcome the limitations of selection bias and explore more in detail the nature and role of compensatory strategies in navigational tasks for older participants. For instance, studies could employ cognitive ethnographic methods to identify compensatory strategies, and then design psychology studies to test the impact of those strategies in navigation performance (for a recent example of cognitive ethnography being employed to study compensatory strategies, see Zubatiy et al., 2023). Future studies should also explore in more detail the influence of task difficulty on real-world validity for older populations in varied navigation tasks.

5. Charts and Tables:

The charts are useful, although not simple to understand they are clearly explained in the legends, effectively supporting the results.

6. References:

The references are comprehensive and relevant to the research. However, some minor improvements could be made.

Suggestions for Improvement:

•In-text Citations: Ensure that all studies mentioned in the discussion are appropriately cited in the references section. There were a few instances where prior work was mentioned without direct references.

Thank you so much. This has been taken care of.

•Reference Formatting: Double-check the consistency of the reference formatting (e.g., author names, journal titles, and volume/issue numbers) to ensure they meet the journal’s style guidelines.

We have double-checked that the reference formatting meets the journal’s requirements.

Reviewer #2: A very interesting study combining navigation testing in a virtual environment versus a real-world navigation task in older adults. The introduction and scientific background are sound and the authors are familiar with relevant literature. I want to note that I am very excited about this research and believe in the value of this topic. Furthermore, open science practices via OSF is applauded. However, regarding statistics and combining the aims and the results, I have some comments.

Thank you for your excitement about this research. We are very grateful for your feedback to strengthen the manuscript.

General comments:

• I notice variations in the terminology used: spatial ability, spatial cognition, navigation, spatial navigation, wayfinding… Is there a way to make this more coherent, or to describe how spatial cognition and navigation are related to each other?

We have gone throughout the document to unify terminology. We now refer primarily to navigation, and only use spatial ability, spatial cognition, etc, when it refers to a broader set of spatial skills.

• When reading the title, I expect to see an analysis on how navigation performance by means of the Sea Hero Quest could predict real-world navigation performance. However, when looking at the text, this is not mentioned. The closest thing appears to be Figure 2, Panel D. I think the study is good and the title is interesting as well, but I am missing the link between both. Please elaborate.

Our title refers to the fact that, if you know from Fig 2D what performance was on level 11 in Sea Hero Quest, you can make some prediction about how that person will perform in our real-world navigation task. This is the same approach taken in our previous article published in PLoS One in 2019 “Virtual Navigation Tested on a Mobile App Is Predictive of Real-World Wayfinding Navigation Performance”. To keep this new research in line with the prior research we adopted a similar title.

Specific comments:

Introduction

• “This is in contrast to studies using object-based spatial tasks, which have mixed results in predicting real-world spatial behaviour (e.g. path integration, wayfinding)”: To make it more clear, can you add an example of object-based spatial tasks?

We have added an example of an object-based spatial task:

This is in contrast to studies using object-based spatial tasks (e.g. the Corsi task to test visuospatial working memory), which have mixed results in predicting real-world behaviour (e.g. path integration, wayfinding) (Malinowski, 2001; Nori, Grandicelli, and Giusberti, 2009; Lippa, Collaer, and Peters, 2010; Garg et al., 2024).

• “Whilst this is important to show ecological validity of an app-based assessment, the extent to which it could predict real-world navigation in later life is unclear due to the difference in navigation ability that occurs with age”: Can you make this more concrete by explaining what difference you expect in navigation ability in older age?

The previous version was overtly vague. We now phrase this as a decline in navigation performance, rather than as a difference in navigation ability:

Whilst this is important to show ecological validity of an app-based assessment, the extent to which it could predict real-world navigation in later life is unclear due to the decline in navigation performance that occurs with age

• “Furthermo

---

## [Decision Letter · Decision Letter 1]

1 Dec 2024

PONE-D-24-38652R1Predicting real-world navigation performance from a virtual navigation task in older adultsPLOS ONE

Dear Dr. Fernandez Velasco,

Thank you for submitting your manuscript to PLOS ONE. After careful consideration, we feel that it has merit but does not fully meet PLOS ONE’s publication criteria as it currently stands. Therefore, we invite you to submit a revised version of the manuscript that addresses the points raised during the review process.

The paper has improved after this first review but still needs further work on aspects that were mentioned in the first set of review but not addressed by the authors.

All reviewer and editor  requests should be addressed before the paper can be accepted.

We look forward to receiving your revised manuscript.

Kind regards,

Sara Eloy, Ph.D

Academic Editor

PLOS ONE

**Additional Editor Comments:**

The paper has improved after this first review but still needs further work on aspects that were mentioned in the first set of review but not addressed by the authors.

The authors should include an extensive statistical explanation about how the research questions were answered by the research, namely whether Sea Hero Quest is predictive of real-world navigation ability in an older population.

Additionally, there is the need to talk about tendencies and not proven facts due to the small size of the experimental subjects involved in the study. A reference for the Navigational Strategy Questionnaire (NSQ) is still missing.

Reviewers' comments:

Reviewer's Responses to Questions

**Comments to the Author**

1. If the authors have adequately addressed your comments raised in a previous round of review and you feel that this manuscript is now acceptable for publication, you may indicate that here to bypass the “Comments to the Author” section, enter your conflict of interest statement in the “Confidential to Editor” section, and submit your "Accept" recommendation.

Reviewer #2: (No Response)

2. Is the manuscript technically sound, and do the data support the conclusions?

Reviewer #2: Yes

3. Has the statistical analysis been performed appropriately and rigorously? 

Reviewer #2: Yes

4. Have the authors made all data underlying the findings in their manuscript fully available?

Reviewer #2: Yes

5. Is the manuscript presented in an intelligible fashion and written in standard English?

Reviewer #2: Yes

6. Review Comments to the Author

Reviewer #2: Thank you for providing a detailed explanation for the reviewer comments. I feel this manuscript has improved a lot. However, there was one comment of my previous review that was not addressed. I feel that this Methods – Statistical analysis section needs to be rewritten profoundly before accepting this paper. I would like to see an extensive statistical paragraph where the research questions and how these will be analyzed are written in detail. For full transparency, I have added my previous feedback below:

Methods – statistical analysis: I miss how you will analyze your research question, namely whether Sea Hero Quest is predictive of real-world navigation ability in an older population. As you are also expecting to find a general decline reflected in a difference between the older group and the younger group of your previous study, this also has to be described in the statistical analysis part how you will test this. If you are including testing for differences on the basis of gender, this also has to be included in the aims at the end of the introduction.

I have no other comments than this one. Thank you for this interesting research and good luck with the publication process.

7. PLOS authors have the option to publish the peer review history of their article (what does this mean? ). If published, this will include your full peer review and any attached files.

**Do you want your identity to be public for this peer review?** For information about this choice, including consent withdrawal, please see our Privacy Policy .

Reviewer #2: **Yes: ** Joyce Bosmans

---

## [Author Response · Author response to Decision Letter 1]

4 Dec 2024

We are grateful to the reviewers for their feedback. We have revised the manuscript in light of these recommendations. Below, we outline how we have addressed the comments by the reviewers. We include the original review comments in black font, our reply in blue font, and quotes from our manuscript in red font. Overall, we have made the following revisions:

Reviewer #2: Thank you for providing a detailed explanation for the reviewer comments. I feel this manuscript has improved a lot. However, there was one comment of my previous review that was not addressed. I feel that this Methods – Statistical analysis section needs to be rewritten profoundly before accepting this paper. I would like to see an extensive statistical paragraph where the research questions and how these will be analyzed are written in detail. For full transparency, I have added my previous feedback below:

Methods – statistical analysis: I miss how you will analyze your research question, namely whether Sea Hero Quest is predictive of real-world navigation ability in an older population. As you are also expecting to find a general decline reflected in a difference between the older group and the younger group of your previous study, this also has to be described in the statistical analysis part how you will test this. If you are including testing for differences on the basis of gender, this also has to be included in the aims at the end of the introduction.

Thanks a lot for bringing this up. It is true that the Statistical analysis section needed to be rewritten. It now reads as follows:

To test whether Sea Hero Quest is predictive of real-world navigation ability, we calculated Pearson’s correlations between real-world wayfinding performance and the virtual performance at each SHQ wayfinding level.

To assess the cognitive decline in the real and the virtual environments, we compared the performance of older versus younger participants. In both environments, we used a chi-square test to compare the proportions of participants in the older versus younger cohorts who completed the task before the time limit.

We also examined whether familiarity with the video game controls might play a role in driving navigation performance. To explore this, we examined whether the training level (Level 1) was correlated with real-world performance for our older group. Since Level 1 is not an assessment of spatial navigation ability but rather video game familiarity / visuo-motor skill (the goal remains visible from the start of the Level), we used Pearson’s correlation analysis to test whether there was a correlation between the distance travelled in Level 1 and the normalised distance in the real world.

---

## [Decision Letter · Decision Letter 2]

13 Dec 2024

PONE-D-24-38652R2Predicting real-world navigation performance from a virtual navigation task in older adultsPLOS ONE

Dear Dr. Fernandez Velasco,

Thank you for submitting your manuscript to PLOS ONE. After careful consideration, we feel that it has merit but does not fully meet PLOS ONE’s publication criteria as it currently stands. Therefore, we invite you to submit a revised version of the manuscript that addresses the points raised during the review process.

The paper has improved after this second review but still one aspect was not addressed. A reference for the Navigational Strategy Questionnaire (NSQ) is still missing. If the following reference was the one used, please include it, otherwise include the correct one.

Zhong, J. Y., & Kozhevnikov, M. (2016). Relating allocentric and egocentric survey-based representations to the self-reported use of a navigation strategy of egocentric spatial updating. Journal of Environmental Psychology, 46, 154-175

We look forward to receiving your revised manuscript.

Kind regards,

Sara Eloy, Ph.D

Academic Editor

PLOS ONE

Journal Requirements:

**Additional Editor Comments:**

The paper has improved after this second review but still one aspect was not addressed. A reference for the Navigational Strategy Questionnaire (NSQ) is still missing. If the following reference was the one used, please include it, otherwise include the correct one.

Zhong, J. Y., & Kozhevnikov, M. (2016). Relating allocentric and egocentric survey-based representations to the self-reported use of a navigation strategy of egocentric spatial updating. Journal of Environmental Psychology, 46, 154-175

Reviewers' comments:

Reviewer's Responses to Questions

**Comments to the Author**

1. If the authors have adequately addressed your comments raised in a previous round of review and you feel that this manuscript is now acceptable for publication, you may indicate that here to bypass the “Comments to the Author” section, enter your conflict of interest statement in the “Confidential to Editor” section, and submit your "Accept" recommendation.

Reviewer #2: All comments have been addressed

2. Is the manuscript technically sound, and do the data support the conclusions?

Reviewer #2: Yes

3. Has the statistical analysis been performed appropriately and rigorously? 

Reviewer #2: Yes

4. Have the authors made all data underlying the findings in their manuscript fully available?

Reviewer #2: Yes

5. Is the manuscript presented in an intelligible fashion and written in standard English?

Reviewer #2: Yes

6. Review Comments to the Author

Reviewer #2: Thank you for your revisions. I hereby agree to accept this manuscript for publication. Good luck with the publishing process.

7. PLOS authors have the option to publish the peer review history of their article (what does this mean? ). If published, this will include your full peer review and any attached files.

**Do you want your identity to be public for this peer review?** For information about this choice, including consent withdrawal, please see our Privacy Policy .

Reviewer #2: **Yes: ** Joyce Bosmans

---

## [Author Response · Author response to Decision Letter 2]

13 Dec 2024

Dear editor, dear reviewers,

Thanks a lot for this further round of review. There was only the following outstanding comment:

The paper has improved after this second review but still one aspect was not addressed. A reference for the Navigational Strategy Questionnaire (NSQ) is still missing. If the following reference was the one used, please include it, otherwise include the correct one.

Zhong, J. Y., & Kozhevnikov, M. (2016). Relating allocentric and egocentric survey-based representations to the self-reported use of a navigation strategy of egocentric spatial updating. Journal of Environmental Psychology, 46, 154-175

We have now included the reference in question for the NSQ into the manuscript. Thanks so much for pointing this out, and thank you for all of the editorial and reviewing work.

---

## [Editor Report · Decision Letter 3]

20 Dec 2024

Predicting real-world navigation performance from a virtual navigation task in older adults

PONE-D-24-38652R3

Dear Dr. Fernandez Velasco,

We’re pleased to inform you that your manuscript has been judged scientifically suitable for publication and will be formally accepted for publication once it meets all outstanding technical requirements.

Kind regards,

Sara Eloy, Ph.D

Academic Editor

PLOS ONE

Additional Editor Comments (optional):

The authors have addressed the reviewers' and editor’s comments.

I believe that the manuscript has the quality to be published at this stage.
---

## [Editor Report · Acceptance letter]

PONE-D-24-38652R3

PLOS ONE

Dear Dr. Fernandez Velasco,

I'm pleased to inform you that your manuscript has been deemed suitable for publication in PLOS ONE. Congratulations! Your manuscript is now being handed over to our production team.

Kind regards,

on behalf of

Professor Sara Eloy

Academic Editor

PLOS ONE